



# Chilean Observation Network De MeteOr Radars (CONDOR): Multi-Static System Configuration & Wind Comparison with Co-located Lidar

Zishun Qiao[1], Alan Z. Liu[1], Gunter Stober[2], Javier Fuentes[3], Fabio Vargas[4], Christian L. Adami[5], and Iain M. Reid[5,6]

[1]Center for Space and Atmospheric Research, Department of Physical Sciences, Embry-Riddle Aeronautical University, Daytona Beach, Florida, USA
[2]Institute of Applied Physics & Oeschger Center for Climate Change Research, Microwave Physics, University of Bern, Bern, Switzerland
[3]European Southern Observatory, Alonso de Cordova 3107, Vitacura, Santiago, Chile
[4]Department of Electrical and Computer Engineering, University of Illinois at Urbana-Champaign, Urbana, Illinois, USA
[5]ATRAD Pty Ltd., Adelaide, Australia
[6]School of Physics, Chemistry and Earth Sciences, University of Adelaide, Adelaide, Australia

**Correspondence:** Alan Z. Liu (liuz2@erau.edu)

**Abstract.** The Chilean Observation Network De MeteOr Radars (CONDOR) commenced deployment in June 2019 and became fully operational in February 2020. It is a multi-static meteor radar system consisting of three ∼1° latitudinally separated stations. The main (central) station is located at the Andes Lidar Observatory (ALO, 30.25°S, 70.74°W) and is used for both transmission and reception. The two remote sites are located to the north and south and are used for reception only. The southern station is located at the Southern Cross Observatory (SCO, 31.20°S, 71.00°W) and the northern station is located at Las Campanas Observatory (LCO, 29.02°S, 70.69°W). The successful deployment and maintenance of CONDOR provide 24/7 measurements of horizontal winds in the mesosphere and lower thermosphere (MLT), and permit the retrieval of spatially resolved horizontal winds, vertical winds, and temperatures. This is possible because of the high meteor detection rates. Over 30,000 quality controlled underdense meteor echoes are detected at the ALO each day and in total ∼88,000 events are detected each day over the three sites. In this paper, we present the system configuration of the CONDOR and discuss the validation and initial results of its data products. The motivations of deploying the CONDOR system also include the combination of results with other co-located ground-based instruments at the ALO, which provide uniquely cross-validated and cross-scale observations of the MLT dynamics with multiple scientific goals.



# 1 Introduction

The upper mesosphere and lower thermosphere is a region of very active atmospheric dynamic processes, with interactions of atmospheric waves from small scale gravity waves to large scale atmospheric tides and planetary waves. These waves and their interactions are key mechanisms driving the variabilities of the mesopause region as well as the thermosphere and ionosphere above. The measurement of neutral atmospheric winds is essential for studying the dynamics in this region. While medium-frequency (MF) and high-frequency (HF) radars were among the earliest radio techniques used for measuring the neutral upper atmosphere (see e.g., Reid, 2015, and references therein), the meteor radar technique has gained significant popularity in the past two decades due to its reliability, easy deployment, small footprint, and significantly improved meteor detection capability (Hocking et al., 2001b; Elford, 2004). Modern specular meteor radars derive neutral winds by detecting Doppler shifts of specular radio echoes from the plasma trails formed during meteor ablation, which appear more or less randomly in space and time in the region between 70 to 110 km altitude. With sufficient accumulation of such detections, a mean neutral wind can be inferred. This technique is proven to be able to provide unbiased wind measurements based on comparison with accurate high resolution Na lidar measurements (Liu et al., 2002; Franke et al., 2005b).

A typical monostatic meteor radar measures horizontal winds at nominally 1 hr temporal and 2 km vertical resolutions. The resolution of meteor radar winds is primarily limited by the number of detected meteors. More meteor detections within a given time interval and spatial volume lead to more accurate and/or higher resolution wind measurements. In recent years, meteor signals have also been explored to infer neutral atmosphere temperature (e.g., Tsutsumi et al., 1994; Hocking, 1999; Yi et al., 2016; Liu et al., 2017) and density (e.g., Stober et al., 2012; Younger et al., 2015; Yi et al., 2018), as well as gravity wave variances and momentum flux (e.g., Hocking, 2005b; Liu et al., 2013; Spargo et al., 2019). All these applications would benefit from increased meteor detections, which contribute to reducing the uncertainties of these difficult-to-acquire quantities (see e.g., Vincent et al., 2010).

The detection rate of meteor radars has increased over the years with improvements in radar peak power, electronics, and detection algorithms. Table 1 lists several selected meteor radar systems around the world, including their transmitting frequency, peak power, and average daily detection of under-dense meteor trails used for wind measurements. The meteor radars are arbitrarily selected to illustrate the dependence of detection rate on the transmitted power and frequency. It is evident that higher power systems are capable of detecting more meteors and systems with similar powers show varying detection counts at different locations, largely due to different operating frequencies and levels of background radio noise. Lower frequency radars can detect more meteors due to the longer wavelength of the radio wave, which is more sensitive to meteor trails with smaller electron line densities as full wave scattering models demonstrate (Poulter and Baggaley, 1977; Stober et al., 2021a, 2023b). A typical 6 kW system at around 35 MHz can detect a few thousands meteors per day, while a 40 kW system can detect about 20,000.

In addition to increasing the transmitting power, an innovative approach to increase meteor detection rates without additional transmitters is the practice of multi-static meteor radar systems. The concept involves using additional receiving antennas to detect both backward and forward scattering from specular meteor echoes. This approach has a very long history, being used





for studies of turbulence (Roper and Elford, 1963; Muller, 1974) and for studies of MLT winds (Deegan et al., 1970). The latter
study indicated the potential of measuring the 2D structure of the horizontal wind using multi-static pulsed radar and attempts
to do that. The technique has also previously applied for meteor orbit determination (see e.g., Jones et al., 2005; Stober and
Chau, 2015; Reid, 2024, and references therein). It was revisited using modern equipment for MLT wind observations by
Stober and Chau (2015), and several other multi-static meteor radars have been developed more recently (e.g., Spargo et al.,
2019; Conte et al., 2021; Yi et al., 2022). This low-cost addition to an existing 'mono-static' system can significantly increase
meteor detections without the full expense of a radar transmitting station. The multi-static meteor radar system introduced
by the current study was designed and developed by ATRAD Pty Ltd (www.atrad.com.au). This system, named the Chilean
Observation Network De MeteOr Radars (CONDOR), consists of a main station at the Andes Lidar Observatory (ALO) in
Cerro Pachón, Chile together with receiving antennas, and two remote stations with receiving-only antennas located 138 km
north and 108 km south of ALO, respectively. The radar system installation began in northern Chile in June 2019, with the
final installation at the northern remote station completed in February 2020, making the radar system fully operational since
then. As listed in Table 1, the average daily meteor detections from the three sites of CONDOR is ∼88,000 in combination,
about 15 to 20 times greater than a typical 6 kW system.

This paper describes the installation of the aforementioned meteor radar system, its initial results, and using neutral atmo-
spheric wind measurements made by the Na wind-temperature lidar at ALO to assess the performance of the winds measured
by this radar. Section 2 details the multi-static system configuration of the CONDOR system. Section 3 summarizes the routine
data products as well as the validation of wind measurements, while Section 4 discusses the potential applications of CONDOR
measurements.

## 2  The multi-static configuration of CONDOR

The geographical layout of CONDOR system is illustrated in Figure 1, with two remote sites located to the north and south
of the main station ALO. All three sites are astronomy observatories with good infrastructure and engineering support, and
have good potential of deploying and operating optical remote sensing instruments together with the meteor radar considering
their excellent seeing conditions (∼300 clear nights per year). The northern site, Las Campanas Observatory (LCO, 29.02°S,
70.69°W, elevation of 2339m), is located in Chile's Atacama Desert and approximately 100 km northeast of the city of La
Serena, inside one of the world's highest and driest regions and not yet affected by light pollution. LCO has a median seeing
values range from 0.6-0.7 arcsec (Thomas-Osip et al., 2008). The southern site, Southern Cross Observatory (SCO, 31.20°S,
71.00°W, elevation of 1140m), is a tourist observatory operated by Municipality of Combarbala in Cerro Peralito in the IV
Region of Chile. Though it is only 3.5 kilometres away from downtown, the light pollution is still negligible. The central site,
Andes Lidar Observatory (ALO, 30.25°S, 70.74°W, elevation of 2520m), is located near the summit of Cerro Pachón and
managed by the Association of Universities for Research in Astronomy (AURA). Multiple upper atmosphere instruments have
been deployed at ALO since 2009, including a sodium (Na) wind/temperature lidar, a mesospheric temperature mapper, and a
all-sky airglow imager (see e.g., Vargas et al., 2022; Liu et al., 2016, and references therein).





| Name | Location | Model | Pulse Code | Frequency | Peak Power | Meteors/day |
|---|---|---|---|---|---|---|
| | Maui, HI[2] | SKiYMET | 3.6 km monopulse | 40.92 MHz | 6 kW | ∼4,000 |
| | Socorro, NM[1] | SKiYMET | 3.6 km monopulse | 35.24 MHz | 6 kW | ∼5,000 |
| | Yellowknife, Canada[1] | SKiYMET | 3.6 km monopulse | 35.65 MHz | 6 kW | ∼2,500 |
| DrAAMER | King George Island[1] | SKiYMET | 1.5 km 7-bit Barker | 36.9 MHz | 30 kW | ∼9,000 |
| SAAMER | Tierra del Fuego, Argentina[1] | SKiYMET | 3.6 km monopulse | 32.55 MHz | 60 kW | ∼14,000 |
| | | | 1.5 km 7-bit Barker after 2019 | | | |
| | Kunming, China[3] | ATRAD | 4-bit complementary | 37.5 MHz | 24 kW | ∼14,000 |
| | Kunming, China[3] | ATRAD | 4-bit complementary | 53.1 MHz | 48 kW | ∼9,000 |
| | Davis Station, Antarctica[4] | ATRAD | 4-bit complementary | 33.2 MHz | 7.5 kW | ∼10,000 |
| | Mohe, China[3] | ATRAD | 4-bit complementary | 38.9 MHz | 24 kW | ∼20,000 |
| | Langfang Observatory, China[5] | ATRAD | 4-bit complementary* | 35.0 MHz* | 48 kW | >40,000 |
| CONDOR | Cerro Pachòn, Chile | ATRAD | 4-bit complementary | 35.15 MHz | 48 kW | ∼35,000 |
| CONDOR | Las Campanas Obs, Chile | ATRAD | 4-bit complementary | 35.15 MHz | Rx only | ∼23,000 |
| CONDOR | Southern Cross Obs., Chile | ATRAD | 4-bit complementary | 35.15 MHz | Rx only | ∼30,000 |

**Table 1.** Various meteor radars around the world, with their transmitter frequency, peak power, and daily detection of under-dense meteors that can be used for wind measurements. [1](Fritts et al., 2012), [2](Franke et al., 2005b), [3](Liu et al., 2017), [4](Holdsworth et al., 2008), [5](Xu et al., 2023), *meteor mode operation of the Langfang dual-frequency ST-meteor radar.

The CONDOR meteor radar was manufactured and installed by ATRAD Pty Ltd, a commercial atmospheric radar company that has built and installed more than thirty meteor radars over the world. The CONDOR transmitter and receiver system was first installed at ALO and the work was completed by June 26, 2019. SCO and LCO are receive-only sites. A receiver-only sys-

tem was installed at SCO next and became operational on July 13, 2019. The receiver-only system at LCO began operation on February 24, 2020. CONDOR uses a high-power folded crossed dipole antenna for transmission, which is designed to provide "all-sky" illumination (Reid et al., 2018). CONDOR also incorporates a GPS disciplined oscillator at each site that provides the GPS locked time and frequency. The receiving antenna array at each site consists of 5 crossed-dipole antennas arranged as an interferometer using either a cross or "T" arrangement, with baseline separations of $2\lambda$ and $2.5\lambda$ in two orthogonal axes on

a plane. Such an array is often called a Jones array (Jones et al., 1998). The receiving array layout at LCO has a "T" shape and ALO and SCO have a cross layout. The operating frequency of CONDOR system is 35.15 MHz, pulse repetition frequency is 430 Hz, range resolution is 1800 m, and the peak transmitting power is 48 kW. Note that the 2.52 km elevation at ALO should be of particular consideration in the geometry to determine the height of detected meteors, compared to the other meteor radars located at near sea level such as the height determination described in Younger (2011). Before all sites became operational,

the ALO meteor radar software was configured to use the mono-static analysis technique, while at SCO the analysis software used the bi-static forward scattering technique. Once all three sites became operational on Feb 24, 2020, the analysis software at ALO was also configured to use the same algorithm for bi-static configuration. For ALO the bi-static algorithm is applied





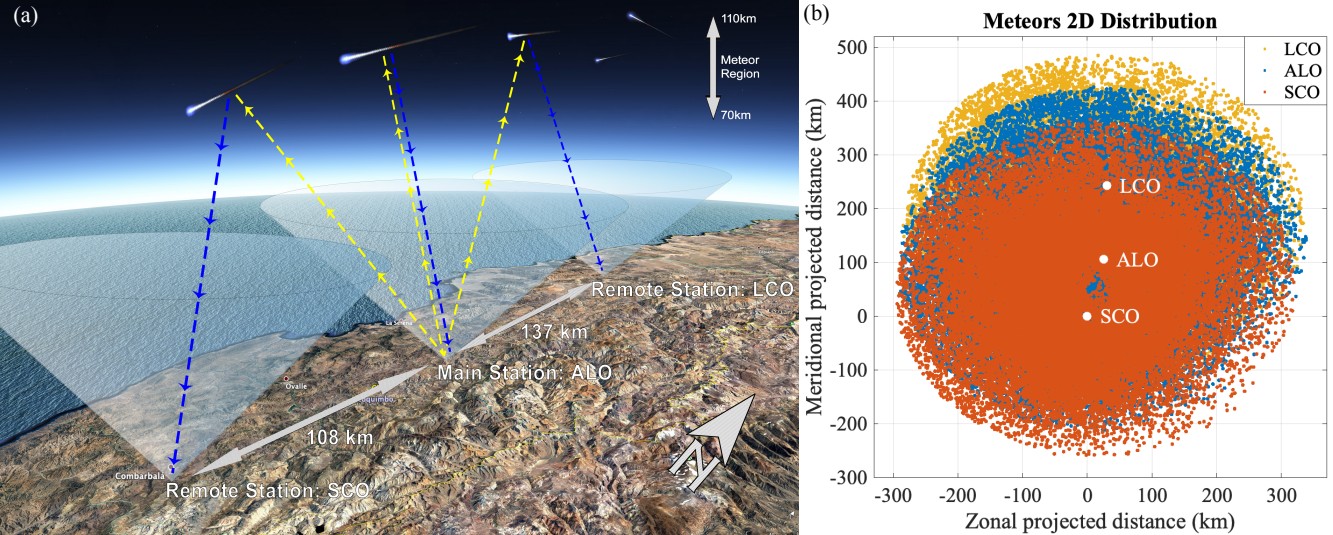

**Figure 1.** (a) Schematic layout of CONDOR multi-static system. The background is adapted from Google Earth over northern Chile. The yellow dashed arrow lines denote the transmitting signals and the blue dashed arrow lines denote the receiving signals and their directions. The cones represent the field of view (FOV) of "good" detections, of which the error codes are zeros. Note that the FOVs are larger and more overlapped at meteor region. (b) The projected 2D distribution of detected "good" meteor events in one day (2020/03/01) at LCO (yellow), ALO (blue), and SCO (red), respectively. The solid white dots denote the relative geo-locations of three sites and x- and y- axes mark the distance from SCO.

with a very small distance between the transmitter and receiver. This provides better consistency in signal processing as well as easier incorporation of geometric factors such as the elevation differences among three sites. The distances from ALO to
SCO and from ALO to LCO are ~108 km and ~137 km, respectively (illustrated in Figure 1a), which are of similar values compared to the ~118 km distance in the MMARIA experiment (Stober and Chau, 2015).

Figure 1b provides the two dimensional projected meteor distributions detected by all three sites during one day, in which the detections are identified as "good" events meeting further data processing criteria. A typical number of quality controlled underdense meteor echoes at ALO is over 30,000 per day and is ~88,000 per day when echoes from three sites are combined.
This is several times larger than a typical mono-static meteor radars (e.g., the radars listed in Table 1) and notably improves the quality of atmospheric parameter estimation. While the software at each of the three sites processes its meteor detections and produces routine data products independently, combining all the echoes in the overlapping volume (as shown in Figure 1b) can reduce the uncertainties of wind estimation (see e.g., Zhong et al., 2021). More sophisticated data processing can also be developed to resolve small-scale structures within the overlapping volume. For instance, Stober et al. (2023a) presents a case
study of identifying volcanic gravity waves utilizing the CONDOR latitudinal keograms, from $\sim 27°$S to $\sim 33°$S. Some of the advanced algorithms for atmospheric parameter estimation are discussed in Section 4.


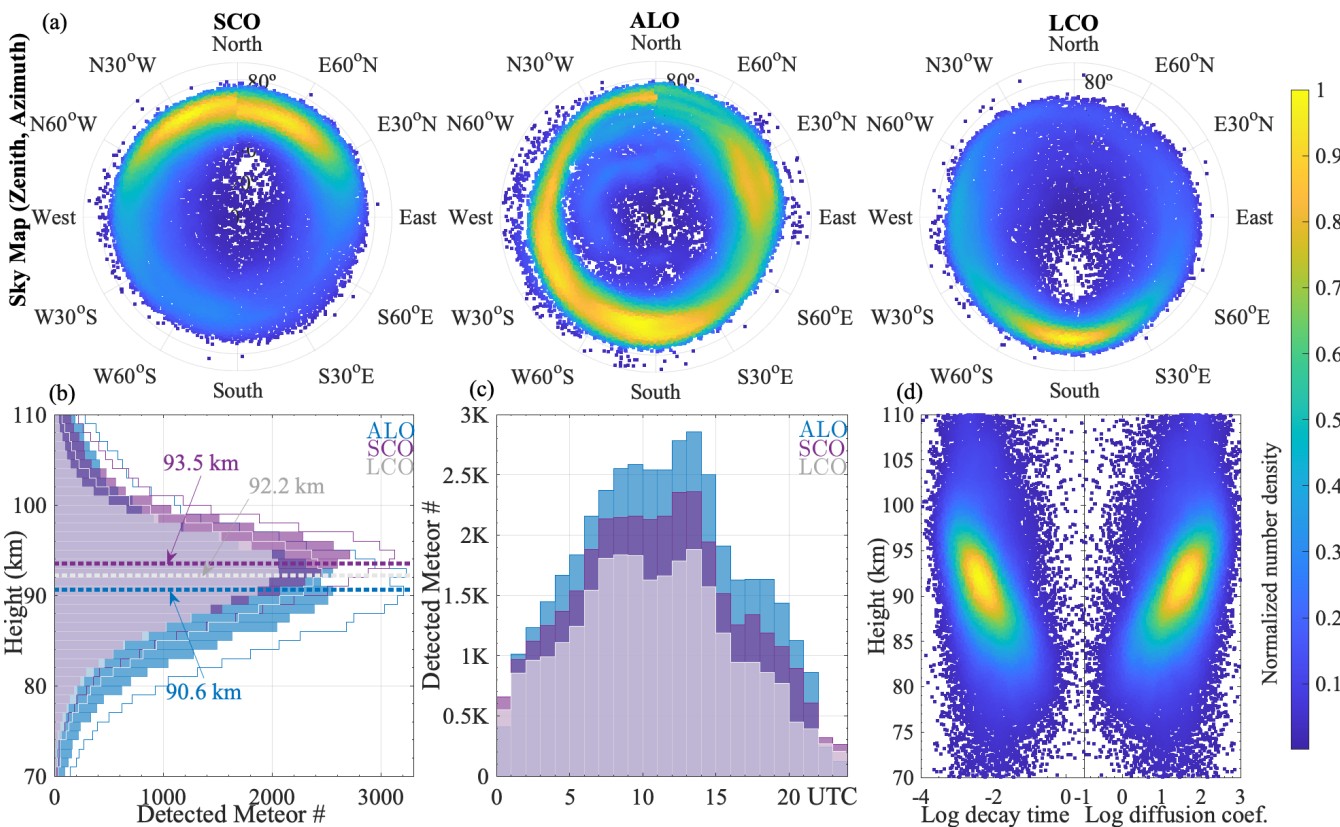

**Figure 2.** Meteor detections in one example day (2020/03/01), plotted as (a) sky maps at three sites, (b) height distribution, (c) universal time distribution. (d) shows the meteor plasma trails decay time and diffusion coefficient. Note that (a) and (d) share the same color map, which represents the normalized number density.

Having introduced the multi-static system layout of CONDOR, we now take an overview of its daily detections. The characteristics of detected meteor echoes are shown in Figure 2 for a one-day (2020/Mar/01). Only the underdense echoes that can be used for wind estimated are included here (Holdsworth et al., 2004) and the overdense signals are discarded from the raw
data (see e.g., Younger, 2011). Figure 2a displays the sky maps of meteor detections at each site. Note that the color presents the normalized number density of detected events, and the discontinuity at north direction is due to the discontinuity of the polar coordinate. The most dense areas of the detections fall between SCO-ALO, and LCO-ALO, which is in between the transmitter and the receivers. To provide a brief idea on the ratio of "good" detections, the non-shaded areas in Figure 2b are meteor detections in total including all error codes while the shaded areas are those of zero error codes. The height distributions at each site is quasi-Gaussian with the peak heights varying between 90 to 95 km. The peak heights at SCO and LCO are both
higher than that at ALO, by 2.9 km and 1.6 km respectively, as presented by the horizontal lines in Figure 2b. The difference in peak height is consistent with the overall height distribution, and is expected in the way that the forwarding scattering peak height is higher than the backward scattering peak height. This is consistent with the results from other forward scattering





meteor radar experiments and is due to the longer effective Bragg wavelength (Stober and Chau, 2015). In addition, the meteor

detection rate has a clear diurnal variation as is observed in all meteor radar locations (Figure 2c). For CONDOR, the detection

rates are highest around 12-13 UT (8-9 LT at local morning) and lowest at around 23 UT (19 LT, local evening). Such local

time dependence has also been reported in other meteor radar measurements (e.g., Yi et al., 2022). Another important measured

parameter is the meteor trail decay time, which is related to the ambipolar diffusion coefficient. Figure 2d shows an example

of the relationship of meteor trail decay time and the ambipolar diffusion coefficient versus height. As will be discussed in

Section 4, this measurement is used to derive layer-averaged temperature.

## 3    Horizontal Winds: Cross-Site Validation & Comparison with Na Lidar Winds

This section discusses the validation and comparison of the routinely derived and most commonly used zonal and meridional

winds. With the meteor radar system running 24 hours per day except some occasional power outages, CONDOR is contin-

uously providing estimations on the zonal and meridional winds as its routine data product, with three temporal resolutions

of 15 min, 30 min, and 1 hour and a vertical resolution of 2 km. Daily counts of valid meteor detections and winds plots

are displayed on the ALO website (http://alo.erau.edu/data/mr/). Correlation analysis of winds measured at three sites is first

performed and its detail is described in 3.1, by which the promising high correlation coefficients confirm the robustness of

wind measurements. We further compare the winds at ALO with the wind measurements of a co-located sodium lidar. While

lidar wind has higher vertical and temporal resolutions but less temporal coverage than meteor radar, considerable consistency

is exhibited between the simultaneously measured winds by these two instruments. Gain factors between the two wind datasets

from radar and lidar are also computed to discuss the different sensitivities to their measured variabilities (i.e., winds), and the

detail is described in 3.2.

### 3.1    Cross-Site Validation of CONDOR Winds

The cross-site validation of CONDOR winds is performed by comparing the simultaneously measured hourly winds from two

of the three sites, utilizing measurements in one example month (2021/01/01-2021/01/31) with a vertical coverage from 70

km to 110 km and resolution of 2km. Scatter plots of the coincident winds at three sites are displayed in Figure 3, in which

the colors denote normalized number density. The majority of zonal winds appear to vary from 50 m/s westward to 100 m/s

eastward (top row), and the meridional winds are mostly in a range of -100 m/s to 100 m/s (bottom row). It is expected that both

of the high dense area (yellow) and the overall distribution (blue) are closely along the red reference line, which is determined

by $y = x$. Correlation coefficients (C.C.s) are also indicated in the plots and have values of ∼0.8-0.9. The correlation coefficient

($\rho$) of two datasets (A,B) is computed as $\rho(A,B) = cov(A,B)/(\sigma_A \sigma_B)$, where $cov(A,B)$ denotes the covariance and $\sigma$ is the

standard deviation. This confirms the overall consistency of wind observations at all sites. Note that the range of zonal winds are

different from that of meridional winds, with the zonal wind dominated by eastward wind at this latitude while the meridional

wind oscillates around zero. The shape or broadness of the scatter distribution pattern is related to the range of winds in general.

Although in January 2021 the simultaneous meridional winds appear to be more highly correlated than the zonal winds, it is





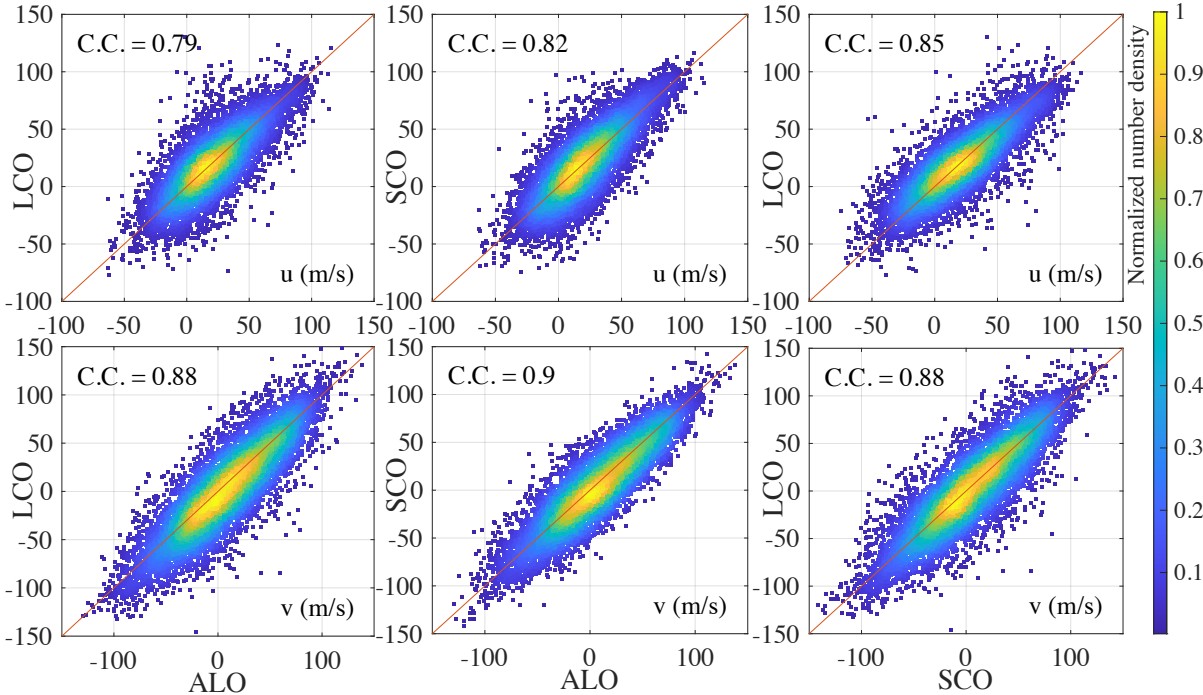

**Figure 3.** Scatter plots of simultaneous hourly zonal (u) and meridional (v) winds measured at ALO, SCO, and LCO, with correlation coefficients (C.C.s) noted in the top-left of each panel. The top row plots the zonal wind comparison and the bottom row presents meridional wind comparison. Red reference lines are $y = x$. The color map denotes normalized number density and data from 2021/01/01 to 2021/01/31 are utilized.

noted that the meridional wind has a more diverse range in each calendar month (not shown) and leads to different values of C.C.s. C.C.s of monthly comparisons for one year are in the range of ∼0.7-0.9, sometimes C.C.(U)>C.C.(V), sometimes vice versa. Seasonal variation of the scatter plot distribution is also noted as the distribution of the meridional winds could have a broader pattern than that of the zonal winds at some months (not shown), which is possibly related to the small-scale

meridionally propagating fluctuations or tidal variations over the ∼1-2° latitudinal difference between three sites.

### 3.2 Radar/Lidar Winds Comparison

With the comparisons of CONDOR hourly winds from all three sites presented above, we now conduct the comparison of winds from meteor radar and a co-located sodium (Na) lidar at ALO. This co-located Na lidar measures the neutral atmosphere wind, temperature, and sodium density at 0.5 km vertical and 6 min temporal resolution and was deployed at ALO since 2009.

Reliability and signal level of this Na lidar has been increased after a system upgrade in 2014, which extends the measurements up to the lower thermosphere (140 km) when the thermospheric Na layer appears (Liu et al., 2016), and provides the capability of detecting turbulence parameters (Guo et al., 2017). Specifically, the neutral wind is measured by detecting the Doppler shift in the sodium $D_{2a}$ line (see e.g., She et al., 1991; Papen et al., 1995; Gardner, 2004) through a three-frequency technique,



taking advantage of the much larger back scatter cross section of sodium atoms available in the mesopause region due to the

meteor ablation. Comparisons of joint dataset of winds from lidar and meteor radar have been conducted by Liu et al. (2002) for

measurements at Starfire Optical Range, New Mexico, and Franke et al. (2005b) for measurements at Maui Space Surveillance

Complex, Hawaii. Since Franke et al. (2005a) attributed the root-mean-square difference in the radar/lidar winds primarily

to the much higher vertical resolution of lidar measurements, it is of particular interest in comparing the high-resolution (15

min×2 km) CONDOR winds with lidar winds, which is notably improved from the previous spatiotemporal resolution of

meteor radar winds of 1 hour×4 km utilized in Franke et al. (2005a) and 1 hour×6 km utilized in Liu et al. (2002).

**Figure 4.** (a) Simultaneous lidar and meteor radar winds on July 27, 2019 at UTC hours 1 to 11. The temporal and spatial resolution of lidar winds are 6 min and 0.5 km, and that of meteor radar winds are 15 min and 2 km. (b) Vertically averaged lidar winds (6 min, 2 km) compared to meteor radar winds (15 min, 2 km) at 88 km height above site, from 2019/07/25 to 2019/07/31. (c) Scatter plot of the coincident lidar/radar winds interpolating to same spatiotemporal resolution of 15 min and 2 km.

After the CONDOR/ALO site became operational in late June 2019, the Na lidar at ALO was operated for several nights

every month in the second half of 2019. The collection of Na lidar data is interrupted since March 2020 due to instrumental

issues and the following pandemic. Coincident lidar measured zonal and meridional winds are available for 47 nights in the



date range of 06/28-07/07, 07/25-08/07, 08/24-09/05, and 09/22-10/05 in year 2019. Such joint measurements are hereby
utilized for the comparison between lidar and CONDOR/ALO winds. Note that before the system changeover of CONDOR
(on 2020/02/24), the height data was recorded as the height above site elevation, and the more commonly accepted height
above sea level (HASL) should take into account an additional 2.52 km at ALO. The vertical coverage of CONDOR/ALO
winds used here is therefore 72.52-112.52 km HASL, and that of the joint lidar winds is 80-105 km HASL. Figure 4a shows
the direct comparison of coincident lidar/radar winds during one example night in 2019/07/27 with their original spatiotemporal
resolutions, i.e., 15 min×2 km for CONDOR winds and 6 min×0.5 km for lidar winds. It appears that the phase, amplitude,
and variation of the large-scale fluctuations (e.g., tides) are captured with overall good agreements in the zonal and meridional
winds of two instruments. However, the lidar winds exhibit more detailed structures than radar winds, such as the two layers
of southward meridional wind at ∼85 km at around 7:00-10:00 UTC in lidar data and the one layer in meteor radar data. For
statistical comparisons with the radar winds we further re-grid the lidar winds to match the spatiotemporal resolution of meteor
radar winds. Figure 4b present the lidar (red) and meteor radar (blue) winds at same vertical resolution but at their original
temporal resolution (6 min and 15 min), and Figure 4c displays the point-to-point comparison of radar and degraded lidar
winds (15 min×2 km). The height of winds in Figure 4b is corresponding to the reference lines (dotted black) in Figure 4a.
Although differences could be noted in Figure 4b particularly relating to small scale variations, the overall tendency of the two
datasets are convincingly robust and consistent. This is further confirmed by the high correlation coefficients, as indicated in
Figure 4c, of which the values are 0.79 of the zonal winds and 0.75 of the meridional winds. It is also noted that the lidar winds
show a consistent trend of having larger value than meteor radar winds in Figure 4c, which is more thoroughly discussed in the
regression analysis below.

Following the method proposed in Hocking et al. (2001a) and Liu et al. (2002), a statistical comparison on the sensitivity of
the co-located lidar and meteor radar wind measurements is performed. This method could be generally applied to two datasets
from two instruments measuring the same physical quantity but have different accuracies to its variabilities. If the datasets of
$\{x_i\}$ and $\{y_i\}$ are written as $x_i = v_i + \delta x_i$ and $y_i = g_0 v_i + \delta y_i$, where $v_i$ and $g_0 v_i$ are the variabilities detected by the two
instruments, respectively, $\delta x_i$ and $\delta y_i$ are the deviations, and the gain factor $g_0$ represents a measure of the relative amplitude
of the variability in two measurements. Using the same notation and assumption of probability density function of $v_i, \delta_x, \delta_y$
described in Hocking et al. (2001a), the aforementioned expression of $x_i$ and $y_i$ could be relating to the gain factor $g_0$ and
the variances of deviations of two measurements ($\sigma_x^2$, $\sigma_y^2$) in the way that $< x_i^2 >=< v_i^2 > + \sigma_x^2$, $< y_i^2 >= g_0^2 < v_i^2 > + \sigma_y^2$,
and $< x_i y_i >= g_0 < v_i^2 >$. Here $<>$ denotes ensemble averages. If we replace $< x_i^2 >$ with the variance of $x_i$ (which is, $s_x^2$),
replace $< y_i^2 >$ with $s_y^2$, and replace $< x_i y_i >$ with the covariance $s_{xy}$, then $g_0$ could be expressed as

$$g_0^2 = (s_y^2 - \sigma_y^2)/(s_x^2 - \sigma_x^2), \tag{1}$$

$$g_0(\sigma_x) = s_{xy}/(s_x^2 - \sigma_x^2). \tag{2}$$

If we divide Eq. 1 by Eq. 2,

$$g_0(\sigma_y) = (s_y^2 - \sigma_y^2)/s_{xy}. \tag{3}$$





Therefore, with computed variance and covariance of the two datasets (i.e., known $s_x^2$, $s_y^2$, and $s_{xy}$), Eq.s 2 and 3 could be understood as the gain factor $g_0$ being a function of only $\sigma_x$ and $\sigma_y$, respectively. Note that the deviations from the "accurate" wind could be a combination of systematic errors and random errors. Although it is assumed that both radar and lidar are

measuring the exact same winds, in reality the fields of view of the two instruments and the processing techniques are different. Hence, the variances of the two datasets not only reflect the intrinsic instrumental error but also contain information about the difference between the actual quantities that two instruments are measuring (Liu et al., 2002).

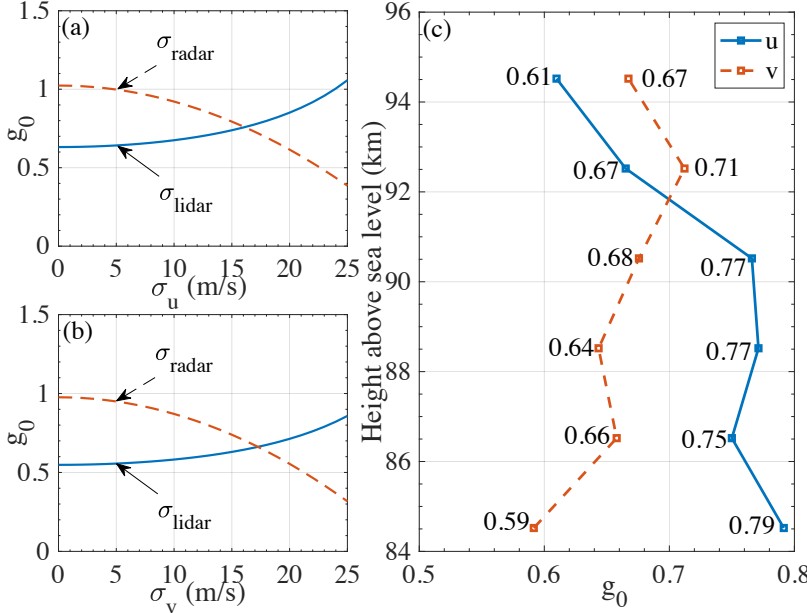

**Figure 5.** (a-b) Gain factor $g_0$ plotted as a function of $\sigma_{radar}$ (dashed red) and $\sigma_{lidar}$ (solid blue) for zonal and meridional winds at all available heights. (c) $g_0$ values at different height for zonal (solid blue) and meridional winds (dashed red), with the known $\sigma_{lidar}$ as the measurement errors. Only over 1,000 joint measurements at each height are considered as sufficient and processed.

We now substitute variables with subscript $x(y)$ in Equations 1-3 with lidar (meteor radar) winds and discuss the relationships of $g_0$ as a function of $\sigma_{lidar}$ or $\sigma_{radar}$, as presented in Figures 5a and b with artificially given $\sigma$ values sampling from

0 m/s to 25 m/s. The crossing points of two curves have $g_0 = 0.76$ for zonal wind and $g_0 = 0.67$ for meridional wind. At $\sigma_u = 0$, $g_{radar} = 1.02$ while $g_{lidar} = 0.63$, and at $\sigma_v = 0$, $g_{radar} = 0.98$ and $g_{lidar} = 0.55$. These numbers are similar to those obtained in Liu et al. (2002). Figure 5c presents the values of $g_0$ at different height with the assumption that the estimated errors of lidar winds could approximately represent $\sigma_{lidar}$, i.e., the values of $g_0$ are computed by revisiting Eq. 2 with known $\sigma_x$. Note that only heights with sufficient joint data points (>1,000) are plotted. As can be seen, the values of $g_0$ appear always less

than 1 for both zonal and meridional winds. This can be interpreted as indicating that the variability of meteor radar winds is generally smaller than that of lidar winds. This observation aligns with the understanding that lidar measurements are confined





to a smaller volume of the atmosphere, rendering them more susceptible to atmospheric perturbations. Meteor radar measured mean winds tend to have smaller variability due to the average over the entire field-of-view volume.

Winds measured by meteor radars have also been compared to that from other co-located instruments, such as medium frequency (MF) radars (e.g., Wilhelm et al., 2017; Reid et al., 2018; Zeng et al., 2022), and Fabry–Perot interferometers (FPIs) (e.g., Lee et al., 2021; Gu et al., 2021). Such comparisons are useful for validating measurements from newly deployed instruments as well as interpreting the relative sensitivity of those measurements. In particular, Reid et al. (2018) indicated that MF radar could underestimate the magnitude of winds to a degree of 10%-20% at 86 km when compared to meteor radar. Both Lee et al. (2021) and Gu et al. (2021) reported that FPI estimated winds are consistently smaller than meteor radar winds. Note that *to what degree* the difference between two measurements depends on the system configurations. It is generally agreed that the large scale perturbations such as tides are well captured by all these instruments, but they have different sensitivity to gravity wave perturbations and turbulence. These results, in combination with the aforementioned radar/lidar comparisons, conclude that lidar shows the most sensitivity compare to both meteor radar and either MF radar or FPI, with meteor radar being more sensitive to MF radar or FPI.

## 4 Discussion on Advanced Data Products

### 4.1 Tomographic reconstruction of 3D wind field

In addition to CONDOR measured horizontal winds, a recently developed algorithm, 3DVAR+DIV has been used to derive the arbitrary horizontal wind field as well as the vertical wind. This algorithm creates a tomographic reconstruction of the 3D wind field based on optimal estimation technique and Bayesian statistics, and has been adapted to CONDOR measurements (Stober et al., 2021c, 2022). The meteor detection rate of the CONDOR is particularly increased by its 48 kW high power transmitter that leads to roughly 30,000 valid detections per day and site. Such a high meteor trail detection rate enables the reconstruction of the 3D wind field within the volume detected by the CONDOR, and controls the uncertainty estimations. Such capability of observing small-scale wind structures, at horizontal scales of tens to hundreds of kilometers is a significant advancement compared to a traditional meteor radar that provides only vertical profiles of horizontal wind with no information on horizontal structure.

Figure 6 shows a comparison of different geographic and cartesian coordinates as well as for very high temporal resolution reaching from 30 minutes to 3 minutes for CONDOR. These retrievals were performed for the analysis of the Hunga Tonga–Hunga Ha′apai (HTHH) volcanic eruption (Stober et al., 2023a). Due to the high meteor detection rate, CONDOR measurements are suitable to benchmark the 3DVAR+DIV retrievals concerning the temporal resolution of the obtained 3D winds while keeping the spatial information. The example shown in Figure 6 is exceptional. We recorded wind speeds above 220 m/s lasting for more than 30 minutes. Such wind speeds have not yet been reported from other observations at that high temporal and spatial resolution on Earth. Furthermore, we demonstrate that temporal resolutions of 5 minutes appear to be feasible with CONDOR for a 30× 30 km grid spacing. We even conducted one retrieval run with a 3-minute temporal resolution and a 40×40 km grid spacing. Our results indicate that the large-scale flow field is well-reproduced down to the 3-minute





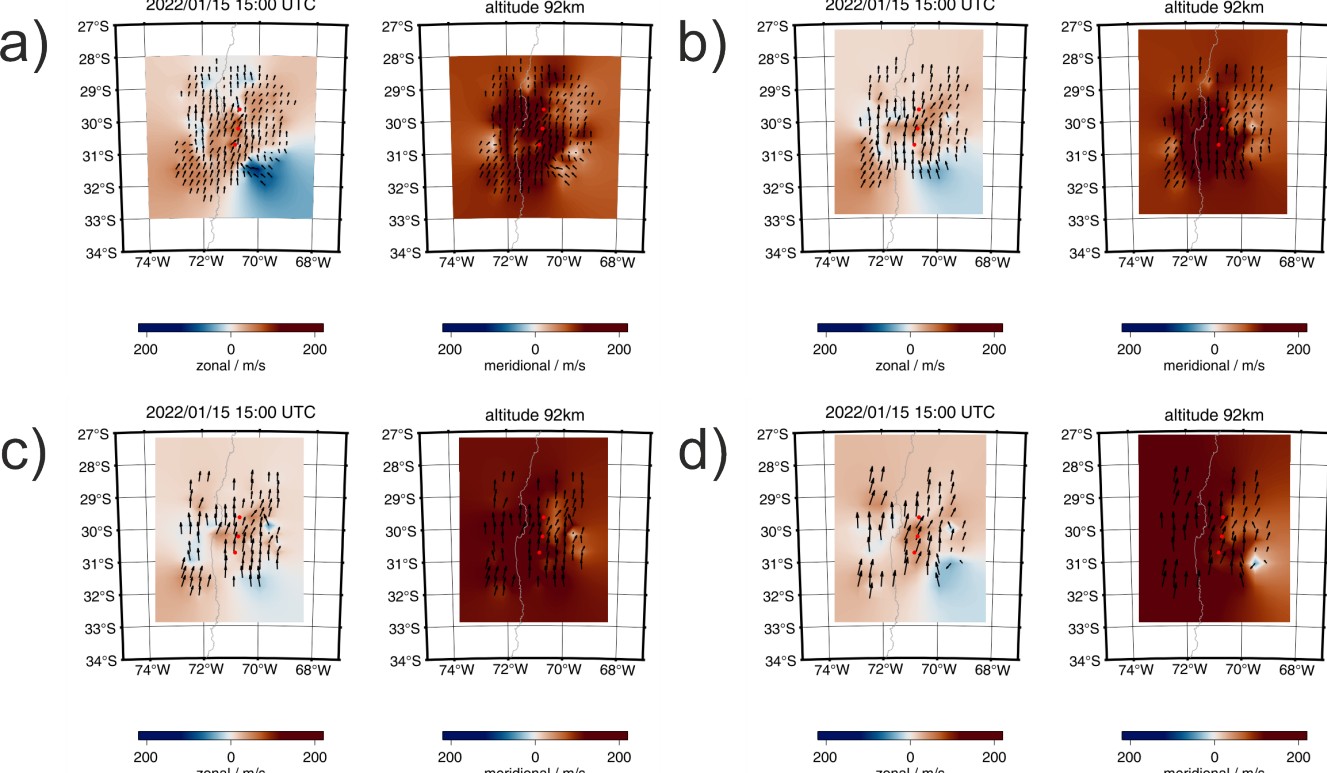

**Figure 6.** Zonal and meridional spatially resolved winds for different temporal and spatial resolutions at 15:00 UTC on 15th January 2022. a) Shows a temporal resolution of 30 min using the geographic grid coordinates with approximately 30×30 km spacing. b)-c) Leverage a cartesian coordinate grid with 30×30 km spacing and temporal resolutions of 10 min and 5 min, respectively. d) Presents a high-resolution retrieval with 3 min temporal resolution applying a 40×40 km cartesian coordinate grid.

temporal resolution, although the problem is much more sparse compared to the 10-minute analysis. This example outlines that the combination of advanced multi-static meteor radar networks such as CONDOR combined with 3DVAR retrievals permits us to reach unprecedented temporal resolutions while sustaining spatially resolved 3D winds.

We further benchmark the capabilities of CONDOR by computing temporal and horizontal wavelength spectra leveraging the high-resolution retrievals from the HTHH analysis. Figure 7 shows the zonal and meridional wind spectra in the time

domain for the 10-, 5-, and 3-minute retrievals, respectively. We added a $k^{-5/3}$-slope for reference. The spectra agree very well for the longer-period waves with periods longer than 1 hour. Only the 3-minute retrievals show a peculiar behavior for the periods between 30 minutes and 1 hour where the slope seems to be less steep, which is likely caused by the sparsity of the inversion problem resulting in relatively larger statistical uncertainties compared to the 10-minute retrieval analysis. For periods below 30 minutes, the slope is again approximately $k^{-5/3}$ before it falls off more rapidly when reaching the spectral resolution

limit. We also calculated horizontal wavelength spectra leveraging all three temporal resolutions. Figure 7 (right panels) shows



**Figure 7.** Temporal and spatial spectra for CONDOR. a), c) and e) show the temporal spectra of the zonal and meridional wind at 92-km altitude for the 10-, 5-, and 3-minute temporal resolution, respectively. These spectra were computed by taking data from 12th-31st January 2022. b), d), and f) visualize zonal horizontal wavelength spectra of the 3D winds for all three temporal resolutions.





the resulting daily mean spectra for the zonal, meridional, and vertical wind based on the 3DVAR+DIV retrieval algorithm. Furthermore, we estimated the spectral slopes for different horizontal wavelength windows. Due to the linear alignment of the passive receiver stations in the North-South direction from the central site at the Andes Lidar Observatory, the zonal domain size is limited and the measurement response for the wind components is not homogeneously distributed over the domain. Zonal winds are most reliable East and West of the the central axis, but show a reduced measurement response directly above the stations, whereas the meridional winds are most reliable along the North-South axis above the sites with a smaller measurement response at the East and West domain boundaries (see Stober et al. (2022), Figure 5).

The GW momentum flux is another key quantity to assess the impact of small-scale waves on large-scale circulation. Meteor radar measurements can be used to calculate the momentum flux, but its uncertainty is high with traditional meteor radars because the second-order moments are more sensitive to measurement uncertainties. CONDOR system significantly enhanced this by detecting many more meteor trails, making it possible to estimate the gravity wave momentum flux at higher temporal resolutions with acceptable uncertainty. GW variances (or wind variances, $\overline{u'^2}, \overline{v'^2}, \overline{w'^2}$) and momentum fluxes ($\overline{u'v'}, \overline{u'w'}, \overline{v'w'}$) contained in the Reynolds stress components (where $u', v', w'$ refers to the fluctuating eastward, northward, and vertical winds) are also computed by the 3DVAR+DIV algorithm. The Reynolds stress components (i.e., GW momentum fluxes and variances) are derived from Reynolds-averaged Navier-Stokes equations based on the method suggested by Hocking (2005a) and the detailed retrieval can be found in Stober et al. (2021b). The momentum flux analysis for CONDOR was implemented based on a general Tikhonov regularization;

$$||Ax - b||^2_P + \lambda||\hat{\Gamma}(x - x_a)||^2_Q \quad . \tag{4}$$

Here the term $||Ax - b||^2_P$ describes the classical least square approach of momentum flux equation as derived in Hocking (2005b) weighted by the statistical uncertainties obtained from a non-linear error propagation denoted by the subscript $P$. However, due to the often limited sampling statistics and all types of measurement errors, we added a generalized Tikhonov constraint. The Tikhonov matrix $\hat{\Gamma}$ is estimated by computing an apriori $x_a$ for a certain time and altitude bin based on the neighbor bins in altitude and time, which is weighted by a covariance $Q$. The relative importance of each term in the cost function is controlled by a Lagrange multiplier $\lambda$. The relative contribution of the Tikhonov depends on the data quality and temporal resolution and can range between 1-10% and for some meteor radars also much higher values. The advantage of this approach is that there are almost no longer negative values of the main diagonal elements and that the solution is more robust and less susceptible to biases and statistical errors caused by the random sampling within a time-altitude bin.

## 4.2 Temperature estimate

A layer-averaged atmospheric temperature in the height range of detected meteors can be related with the ambipolar diffusion coefficient of the meteor plasma trails (Tsutsumi et al., 1994; Hocking et al., 1997; Hocking, 1999)

$$T = S \cdot \log_{10} e \cdot \left( 2\frac{dT}{dz} + \frac{mg}{k} \right), \tag{5}$$

where $T$ is the estimated layer temperature; $e$ is the Euler's number; $m$ is the molecular weight of the air; $g$ is the gravity acceleration and $k$ is the Boltzmann constant. $dT/dz$ is the average vertical temperature gradient in the layer and $S$ is the slope





of a linear model of $z = S \cdot \log_{10} D_a + c$, where $z$ is height, $D_a$ is the measured ambipolar diffusion coefficient of meteor trail
plasma and $c$ is a constant. Only the underdense meteor detections are used for this estimate (e.g., Kaiser and Closs, 1952) and
the potential polarization effects could be mitigated by utilizing detections with zenith angle less than a certain value. There
are different practices to obtain $dT/dz$ and $S$. For example, Stober et al. (2008) presented a way to adjust the temperature
gradient model based on TIMED/SABER measurements. Yi et al. (2016) combined the TIMED/SABER, Aura/MLS, and a
global temperature gradient model from Hocking (1999) to fit the annual, semi-annual, and ter-annual components of their used
temperature gradients. As a practice for CONDOR temperature derivations, Qiao et al. (2024) calculated the layer temperature
using $dT/dz$ based on temperature profiles from MSIS (Hedin, 1991). The outlier effects in linear regression could also be
reduced in many ways. For example, Qiao et al. (2024) limits the probability density from kernel density estimation and
performs a robust regression. The temporal resolution of the estimated temperatures depends on the detection rates. Reliable
temperature can be obtained at daily resolution at CONDOR. It is therefore suitable for resolving large-scale oscillations such
as planetary waves (Stober et al., 2012).

## 5 Conclusions

The successful deployment and maintenance of CONDOR provide continuous measurements of horizontal winds in the MLT
region and facilitate the retrieval of spatially resolved horizontal wind fields, vertical winds, and temperatures. This is partic-
ularly benefited by its extensive detections, amounting to over 30,000 quality controlled underdense meteor echoes at ALO
and approximately 88,000 events per day in total. This paper presents the system configuration of CONDOR and discusses the
validation and initial results of its data products. Specifically, several advantages of this multi-static meteor radar system are
listed below:

1. The increased detection of meteors is achieved at relatively low costs, making the estimations of atmospheric parameters
more reliable and at higher resolutions.

2. The spatial coverage is significantly extended by the multi-static configuration, with receiving stations as far as 250 km from
the transmitter still able to detect meteor trails.

3. The CONDOR routine winds are not only highly correlated with but also point-to-point comparable to the lidar winds,
providing robust and continuous wind measurements to the research community.

4. Detection of the same volume from different directions allows for better estimation of the wind vector. Even if the number of
meteor detections is large enough for tomographic reconstruction, wind rotation (vorticity) cannot be estimated from a mono-
static system because all measured Doppler shifts are relative to the same location, thus only the divergent field is included in
the estimated wind field. The multi-static system removes this limitation of monostatic systems and allows for the estimation
of wind rotation.

We conclude that CONDOR, in combination with other co-located ground-based instruments at ALO, provides uniquely
cross-validated and cross-scale observations of the MLT dynamics.



*Data availability.* CONDOR winds with the HDF5 format are available at the NSF CEDAR madrigal database. To access the data, follow these steps: select 'Access Data', then 'List Experiments', choose 'Meteor Radars' from the list of 'Choose instrument category(s)' and find 'CONDOR multi-static meteor radar system' under 'Choose instrument(s)'.

*Author contributions.* ZQ prepared the manuscript with contributions from AL, GS, and JF; ZQ and GS performed the data analysis; AL led the CONDOR installation with contributions from JF, FV, and CA; All authors reviewed the manuscript.

*Competing interests.* The authors declare no conflict of interest.

*Acknowledgements.* Z. Qiao and A. Z. Liu acknowledge the excellent support provided by the Cerro Pachón astronomy facility managed by the Association of Universities for Research in Astronomy, the Las Campanas Observatory, and the Southern Cross Observatory. Z. Qiao acknowledges the discussions with Wen Yi, Xianghui Xue, and Jorge (Koki) Chau. The meteor radar acquisition is funded by the National Science Foundation (NSF) Major Research Instrumentation grant AGS-1828589. The Na lidar operation at ALO is being supported by the NSF grants AGS-1759471 and AGS-1759573. The involvement of A. Z. Liu and G. Stober were also supported by the International Space Science Institute (ISSI) in Bern, through ISSI International Team project #23-580 "Meteors and phenomena at the boundary between Earth's atmosphere and outer space". G. Stober is a member of the Oeschger Center for Climate Change Research (OCCR). The 3DVAR+DIV retrievals were developed as part of the ARISE design study (http://arise-project.eu/, last access: 8 October 2020) funded by the European Union's Seventh Framework Programme for Research and Technological Development.



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
