# Peer review of "Chilean Observation Network De MeteOr Radars (CONDOR): Multi-Static System Configuration & Wind Comparison with Co-located Lidar"

_Atmospheric Measurement Techniques, 2024_

## Referee Comment (RC1)

This manuscript introduces an interesting multi-static meteor radar system, Chilean Observation Network De MeteOr Radars (CONDOR), and its wind comparison with co-located lidar. The initial observation results demonstrate that this new CONDOR system has excellent meteor detecting capability, therefore obtains higher time resolution (such as 15 min) and precise winds comparing with co-located lidar. Interesting 3D wind fields and horizontal wavelength spectra during Hunga Tonga–Hunga Ha′apai (HTHH) volcanic eruption are also exhibited and exceptional 220 m/s wind speed are recorded at the same time. This manuscript is worth publishing, but I still have some comments.

1. Line 75: the seeing values range of LCO site could be introduced in section 3.2 when the authors describe the ALO lidar system. Seeing value seems more related with optical observations and can explain how the lidar obtains such high-quality wind data that are used to compare with CONDOR.

2. I wonder why the authors give the discontinuous polar diagrams in Figure 2 (a)? Complete polar diagrams might give more information and better explanation of the meteor distribution.

3. Line 278: the authors describe the GW momentum fluxes algorithm developed for multi-static meteor radar system but no results are followed. I guess it's the author's future work and can be briefly introduced in last section?

4. Section 4.2: I recommend the authors remove this section.

---

## Author Comment (AC1)

**Response to reviewers' comments on** *Chilean Observation Network De MeteOr Radars (CONDOR): Multi-Static System Configuration* & *Wind Comparison with Co-located Lidar*

We thank the editor for the opportunity to revise the manuscript AMT-2024-126 for publication in the *Atmospheric Measurement Techniques*. We appreciate the time and effort that the editor and reviewers invested in providing constructive and encouraging feedback. We have incorporated most of the suggestions made by the reviewers and those changes are highlighted in the manuscript.

**Response to the Reviewer 1**

*Thank you for submitting your manuscript regarding a radar network for retrieving horizontal wind profiles. I think the work is interesting, and I have only some technical comments that could help you to improve some points of your work:*

We appreciate the reviewer's encouraging feedback and the valuable comments and suggestions. The comments have been addressed in the point-to-point response below.

• *1. The most relevant for me is the use of the term radar. I know that the name is correct, but I think it is more referred to weather radar detecting echoes in 3D volumes. Maybe it would be more appropriate to use the term bidirectional radar.*

We indeed used the broader term 'radar' several times to refer to the meteor radar (our system or in general) in the text, and the reviewer is right that this could be misleading. To address this comment, we have replaced the term 'radar' with 'meteor radar' (when referring to meteor radars in general) or 'CONDOR' (when referring specifically to our system) throughout the main text to avoid potential confusion. These changes can be found in lines 36, 42, 55, 59, 60, 65 of section 1; lines 105, 152 of section 2; lines 172, 186, 199, 204, 206, 229, and 253 of section 3 in the tracked-change manuscript.

• *2. In table 1, the first column indicates a network, isn't it? Then, I suggest changing the column name to "Radar network" or something similar.*

All listed meteor radars are monostatic systems except for the CONDOR stations, and only the stations with both transmitter and receiver are noted with values of peak powers. The first column of table 1 is intended to list the acronym names of some meteor radars which are designed to be memorable. For example, DrAAMER represents the Drake Antarctic Agile MEteor Radar and the Southern Argentina Agile MEteor Radar is shorten as SAAMER. The other meteor radars without acronym names are generally referred to by their locations (in the second column), such as the Mohe meteor radar, the Davis meteor radar, etc. To clarify, we've changed "name" to "acronym" in table 1.

Since Table 1 aims to highlight the relationship between meteor radar detection rate

and transmitting power and frequency, the choice of meteor radars are rather arbitrarily than a complete global survey (which we prefer to avoid doing so due to the large amount and unfamiliarity of all meteor radars), as described in lines 37-45 in the main text of the tracked-change file. We have added some explanations in the caption of Table 1 to enhance readability.

• *3. Finally, I missed some comparisons of your results with other similar networks, such as the ones presented in Table 1.*

We thank the reviewer for this helpful comment. We have added a paragraph in the text comparing CONDOR with three other multi-static meteor radar networks. These meteor radar networks are properly referenced and comparable to CONDOR, but were not added to Table 1 due to some information missing. The added discussions are focusing on comparing the meteor detection rates (daily counts and distributions), while the winds at different locations are not directly comparable. Detailed changes can be found in lines 123-141 in the tracked-changes file.

**Response to the Reviewer 2**

*This manuscript introduces an interesting multi-static meteor radar system, Chilean Observation Network De MeteOr Radars (CONDOR), and its wind comparison with co-located lidar. The initial observation results demonstrate that this new CONDOR system has excellent meteor detecting capability, therefore obtains higher time resolution (such as 15 min) and precise winds comparing with co-located lidar. Interesting 3D wind fields and horizontal wavelength spectra during Hunga Tonga-Hunga Ha'apai (HTHH) volcanic eruption are also exhibited and exceptional 220 m/s wind speed are recorded at the same time. This manuscript is worth publishing, but I still have some comments.*

We appreciate the reviewer's encouraging feedback and the valuable comments and suggestions. The comments have been addressed in the point-to-point response below.

• *1. Line 75: the seeing values range of LCO site could be introduced in section 3.2 when the authors describe the ALO lidar system. Seeing value seems more related with optical observations and can explain how the lidar obtains such high-quality wind data that are used to compare with CONDOR.*

We thank the reviewer for this constructive comment. The seeing values of LCO are moved to section 3.2 and some related description has been added in lines 174-177 of the tracked-change file.

• *2. I wonder why the authors give the discontinuous polar diagrams in Figure 2(a)? Complete polar diagrams might give more information and better explanation of the meteor distribution.*

We have corrected the discontinuity in the polar diagrams and our apologies for the possible confusion.

• *3. Line 278: the authors describe the GW momentum fluxes algorithm developed for multistatic meteor radar system but no results are followed. I guess it's the author's future work and can be briefly introduced in last section?*

We thank the reviewer for this valuable comment. We included a discussion on GW momentum fluxes to touch on some potentials of CONDOR measurements, which, while not directly presented in this manuscript, have been scientifically validated. To improve and clarify this point, the last two paragraph of section 4.1 has been rewritten in the way of only focusing on the methods and results that have been previously published (lines 300-319 in the tracked-change file). This is indeed an ongoing effort, and we hope the revised text clarifies that the discussion on GW momentum flux retrievals based on CONDOR measurements is intended as a brief review.

• *4. Section 4.2: I recommend the authors remove this section.*

The discussion of temperature estimation (section 4.2) has been removed.

---

## Referee Report (RR1)

This manuscript introduces an interesting multi-static meteor radar system, Chilean Observation Network De MeteOr Radars (CONDOR), including its installation, initial results and wind comparison with co-located lidar. This manuscript is worth publishing. I just have one more suggestion here.

The authors claimed that "The successful deployment…permit the retrieval of …vertical winds." But vertical winds are not shown in section 3 or 4. Could the authors supplement the results of vertical wind?

---

## Author Response (AR2)

**Response to reviewers' comments on** *Chilean Observation Network De MeteOr Radars (CONDOR): Multi-Static System Configuration & Wind Comparison with Co-located Lidar*

We thank the editor for the opportunity to revise the manuscript AMT-2024-126 for publication in the *Atmospheric Measurement Techniques*. We appreciate the time and effort that the editor and reviewers have invested in providing helpful feedback. We have incorporated the suggestion made by the reviewer, and these changes are highlighted in the manuscript.

**Response to Reviewer's Round 2 Comment**

*This manuscript introduces an interesting multi-static meteor radar system, Chilean Observation Network De MeteOr Radars (CONDOR), including its installation, initial results and wind comparison with co-located lidar. This manuscript is worth publishing. I just have one more suggestion here.*

• *The authors claimed that "The successful deployment...permit the retrieval of ...vertical winds." But vertical winds are not shown in section 3 or 4. Could the authors supplement the results of vertical wind?*

We appreciate the reviewer's encouraging feedback and constructive suggestion. The vertical wind results have been supplemented and are now described in the text (lines 273-275).